# Non-Overlapped Multi-View Weak-Label Learning Guided by Multiple Correlations

## ABSTRACT

Insufficient labeled training samples pose a critical challenge in multi-label classification, potentially leading to overfitting of the model. This paper delineates a criterion for establishing a common domain among different datasets, whereby datasets sharing analogous object descriptions and label structures are considered part of the same field. Integrating samples from disparate datasets within this shared field for training purposes effectively mitigates overfitting and enhances model accuracy. Motivated by this approach, we introduce a novel method for multi-label classification termed Non-Overlapped Multi-View Weak-Label Learning Guided by Multiple Correlations (NOMWM). Our method strategically amalgamates samples from diverse datasets within the shared field to enrich the training dataset. Furthermore, we project samples from various datasets onto a unified subspace to facilitate learning in a consistent latent space. Additionally, we address the challenge of weak labels stemming from incomplete label overlaps across datasets. Leveraging weak-label indicator matrices and label correlation mining techniques, we effectively mitigate the impact of weak labels. Extensive experimentation on multiple benchmark datasets validates the efficacy of our method, demonstrating clear improvements over existing state-of-the-art approaches.

## KEYWORDS

multi-label classification, sample augmentation, weak-label learning, label correlations

## 1 INTRODUCTION

Multi-label learning stands as a prominent area within machine learning and pattern recognition. This paradigm involves representing each sample with a feature vector while allowing for the simultaneous association of multiple category labels. The overarching objective revolves around inducing a function capable of assigning multiple appropriate labels, drawn from a predefined label set, to unseen instances [30, 34]. The introduction of multi-label learning has spurred significant scholarly interest, leading to the development of numerous efficient algorithms.

However, a common challenge arises in scenarios where the available number of labeled samples proves insufficient for training an effective multi-label classification model. Addressing this

*MM '24, 28 October - 1 November 2024, Melbourne, Australia*
© 2024 Association for Computing Machinery.
ACM ISBN 978-x-xxxx-xxxx-x/YY/MM...$15.00
https://doi.org/10.1145/nnnnnnn.nnnnnnn

limitation becomes imperative in the quest to enhance model accuracy. In practical scenarios, datasets often describe akin objects with overlapping label sets, thereby forming what we term as a 'same field' within multi-label data. For instance, consider the multi-label image datasets portraying natural scenes depicted in Figure 1. These datasets frequently feature analogous objects such as 'beach','sky','ocaen' and 'water', and exhibit comparable label sets. Leveraging datasets within the same field for joint training holds the promise of significantly augmenting the sample pool, thereby effectively mitigating the challenge of insufficient training samples-a common precursor to overfitting in multi-label classification models. This strategy effectively embodies a Non-Overlapped Multi-View Weak-Label Learning framework (sample augmentation multi-label framework), offering a viable solution to counteract overfitting resulting from limited training samples.

The Non-Overlapped Multi-View Weak-Label Learning framework encounters two primary challenges. Firstly, samples from distinct datasets often exhibit disparate feature types and lack intersection, as depicted in Figure 2(c). Consequently, conventional multi-view multi-label classification approaches [22] are unsuitable for direct application. Secondly, corresponding label sets across datasets may not align precisely. For instance, in Figure 1, data set $A$ includes the label 'clouds,' which is absent in data set $B$. Thus, when jointly training these datasets, 'clouds' becomes a missing label for images in data set $B$ featuring clouds, exacerbating the weak-label problem. Weak-label learning primarily addresses multi-label learning scenarios with partially relevant label sets [24] [13] [4].

To tackle these challenges, researchers typically explore multi-view and weak-label learning strategies. However, existing methodologies struggle to effectively address both challenges simultaneously. Notably, two frameworks closely related to our problem have been proposed in prior works (refer to Figure 2). The multi-view multi-label framework [22, 35] (depicted in Figure 2(a)) concurrently handles multi-view and multi-label classification. Building upon this framework, Tan *et al.* [25] introduced an extension termed incomplete multi-view multi-label classification (illustrated in Figure 2(b)), focusing on addressing incomplete views and missing labels concurrently [29] [26] [15] [16]. Nevertheless, these frameworks encounter difficulty in addressing the two challenges posed by sample augmentation multi-label learning (illustrated in Figure 2(c)). A comparison among the three sub-figures (a, b, c) in Figure 2 reveals that our proposed Non-Overlapped Multi-View Weak-Label Learning Guided by Multiple Correlations (NOMWM) offers a more general solution compared to the aforementioned frameworks.

To this end, we designed our model to specifically address the aforementioned challenges. Firstly, to leverage samples with diverse features comprehensively, we adopt a multi-view learning approach to jointly handle datasets with varying features. By learning a common subspace, we fuse information from different datasets

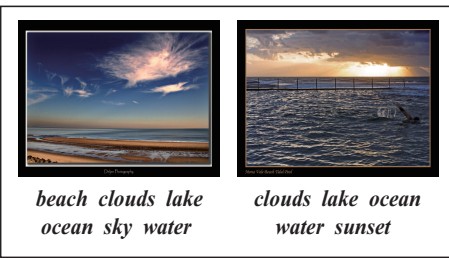

| *beach clouds lake ocean sky water* | *clouds lake ocean water sunset* |
| *beach sky sea tree sunset hill man palm* | *beach ocean surf water* |

*Some examples in data set A*                    *Some examples in data set B*

**Figure 1: Examples of multi-label data sets that belong to the 'same field'. In this paper, we define that data sets belong to the 'same field' if they describe similar objects and have similar label sets. Data set _A_ and data set _B_ in Figure 1 both describe natural scenes, and they have similar label sets. We consider such two data sets belong to the 'same field' in this paper. The samples of these two data sets do not overlap, and their label sets partially overlap but are not completely consistent.**

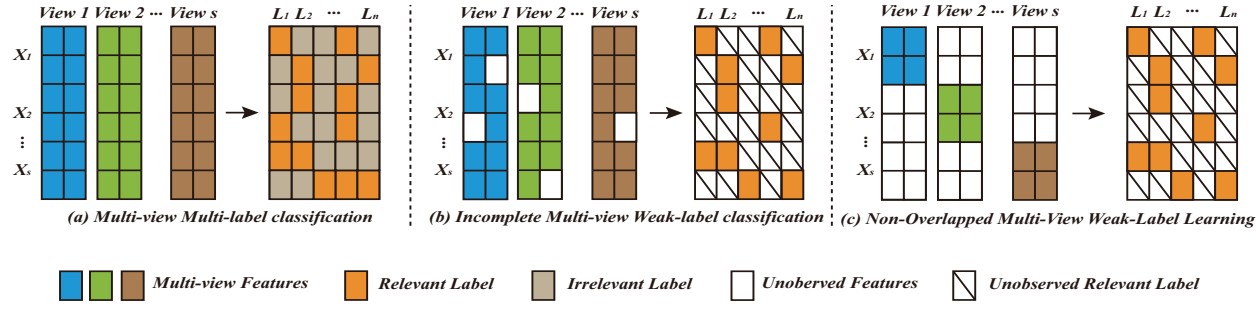

**Figure 2: The proposed Non-Overlapped Multi-View Weak-Label Learning Guided by Multiple Correlations (in sub-figure(c)) is more challenging to deal with compared with the existing two famous frameworks (in sub-figure(a) and sub-figure(b)).**

to enhance model training collectively. Secondly, to contend with inconsistent label sets, we frame the issue as a weak-label learning problem. Specifically, we employ a missing label indicator matrix and label correlation mining techniques. The missing label indicator matrix mitigates the impact of missing labels on the model, while label correlation mining enhances prediction accuracy from a label space perspective. Integration of these aspects facilitates effective sample augmentation for training a more robust multi-label classification model.

In summary, the main contributions of this paper are delineated as follows:

(1) Proposing a novel multi-label learning framework facilitating effective sample augmentation to alleviate overfitting arising from insufficient training samples in multi-label classification. This framework integrates samples from diverse-featured datasets within the 'same field' and addresses weak-label issues stemming from label set mismatches between datasets.

(2) Introducing a novel common subspace learning method to derive a unified representation from disparate training datasets. This method leverages multi-view learning to enhance model prediction accuracy.

(3) Introducing a weak-label learning module to tackle label set inconsistencies across datasets. This module incorporates a label indicator matrix and label correlation mining techniques to mitigate the impact of missing labels and enhance framework robustness.

## 2 RELATED WORK

The landscape of existing multi-label learning algorithms [[30], [34]] broadly divides into two categories: *problem transformation methods* (PTMs) and *algorithm adaptation methods* (AAMs). Despite their differences, both PTMs and AAMs grapple with a common challenge in multi-label learning tasks-namely, the scarcity of instances featuring identical features and complete label sets for training classification models. Frequently, practitioners can only access diverse datasets describing similar objects with comparable label sets. Consequently, the crucial issue becomes how to amalgamate instances with differing features and incomplete label sets during the learning process. Our work closely relates to two key aspects: weak-label learning and multi-view multi-label learning, which we review as follows.

In weak-label learning, a large number of samples have incomplete labels, focusing on multi-label learning scenarios with partially relevant label sets. Noteworthy methodologies include WELL proposed by Sun *et al.* [24], which employs a low-rank similarity matrix hypothesis to uncover instance correlations and complete label dissemination. DM2L by Chen *et al.* [33] leverages the low-rank structure of the label matrix to integrate global and local low-rank label structures alongside discriminative label information. KWLRL by Zhao [32] reconstructs the label semantic space through joint label correlation, leveraging label information consistency and feature-label dependency assumptions, which also

extends the linear model to a nonlinear kernel method to handle data separability challenges. Regarding multi-view multi-label learning, LVSL by Zhao *et al.* [31] tackles non-alignment issues among views and labels through a unified framework, leveraging global and local structural information and view contribution weights. Similarly, NAIM³L by Li *et al.* [11] addresses missing labels, incomplete views, and non-alignment based on global and local label structures, introducing an indicator matrix for handling missing labels and aligning labels in a common space. AIMNet by Liu [14] proposed an attention-driven technique to address the widespread incompleteness issue in multi-view features and labels, which also utilizing statistical weak label correlations and graph attention networks to improve classification accuracy.

While the aforementioned methods are closely related to our work, they predominantly address weak-label learning and multi-view multi-label learning separately. Directly applying these methods to multi-label learning tasks involving multiple datasets with differing features and incongruent label sets proves challenging. Specifically, we face two primary challenges when addressing sample augmentation in these tasks: (1) effectively leveraging multiple datasets with diverse features lacking sample overlap, and (2) addressing inconsistencies in label sets among datasets involved in sample augmentation. To tackle these challenges concurrently, we propose our method, Non-Overlapped Multi-View Weak-Label Learning Guided by Multiple Correlations (NOMWM). Our approach seamlessly integrates datasets with varying features within the 'same field' to learn a multi-label classification model while adeptly handling the issue of incomplete label set matching across multiple datasets. Further details are elaborated in the subsequent section.

## 3 PROPOSED APPROACH

For multi-label classification, let $D = \{(x_i, y_i)\}_{i=1}^{m} = \{X, Y\}$ represent a collection of $d$-dimensional training instances $X \in \mathbb{R}^{m \times d}$ paired with their associated labels $Y \in \{0, 1\}^{m \times n}$, where $m$ and $n$ denote the number of instances and label attributes, respectively. The primary objective of multi-label classification algorithms is to train a predictor $f : X \rightarrow Y$ from $D$ during the training stage, enabling the prediction of the label $\hat{y}$ for a given test instance $\hat{x}$.

Our proposed Non-Overlapped Multi-View Weak-Label Learning Guided by Multiple Correlations (NOMWM) aims to develop a novel multi-label classification model capable of effectively handling datasets with disparate features within the 'same field' to facilitate sample augmentation. Moreover, the model adeptly addresses the weak-label problem, efficiently managing the issue of incomplete label set matching across multiple datasets. The NOMWM model comprises two core components: latent space representation learning (Figure 3) and weak-label classification model (Figure 4).

We leverage the concept of multi-view learning to jointly train multiple datasets featuring distinct features and extract informative patterns from the samples. While traditional multi-view learning primarily focuses on amalgamating information from diverse views within a single dataset, NOMWM extends this concept to integrate different datasets lacking intersection, thus achieving sample augmentation.

Within this framework, we denote $X_v \in \mathbb{R}^{m_v \times d_v}$ as the training datasets with the $v$th feature vector, $Y_v \in \{0, 1\}^{m_v \times n_v}$ as the corresponding label sets, $V_v \in \mathbb{R}^{m_v \times d_c}$ and $U_v \in \mathbb{R}^{d_c \times d_v}$ as the latent space representation and transformation matrix, respectively. $W$ represents the mapping matrix from the latent space to the label space, $S_v$ indicates the similarity between instances in the $v$th training dataset, and $L$ denotes the similarity between labels. Given the need to amalgamate diverse datasets for learning, we first unify the label sets. Herein, $Y \in \{0, 1\}^{m \times n}$ represents the integrated label set. However, due to label set expansion, certain labels present in the original label set $Y_v \in \{0, 1\}^{m_v \times n_v}$ may become missing labels in the new label set $Y \in \{0, 1\}^{m \times n}$. Thus, we introduce $M$ as the indicator matrix to denote missing labels.

Then the objective function of our proposed NOMWM can be written as follows.

$$\min_{V_v, U_v, W, L} \quad \Phi(X_v, V_v, U_v) + \Psi(V_v, W, Y, S_v, L) \tag{1}$$

where $\Phi(X_v, V_v, U_v)$ and $\Psi(V_v, W, Y, S_v, L)$ denote the losses of latent space representation learning (Figure 3(a)) and weak-label classification with label correlation (Figure 3(b)).

### 3.1 Latent space representation learning

In the realm of latent space representation learning, our approach entails projecting datasets $X_1, X_2, \ldots, X_s$ originating from diverse feature spaces into a unified common subspace denoted as $V$, while preserving the consistency of feature magnitudes across samples within this shared subspace (depicted in Figure 3(a)).

Observing Figure 2(c), it becomes apparent that integrating insights from multiple disparate datasets is imperative for joint learning. These datasets feature non-overlapping training data and exhibit distinct feature characteristics. Drawing inspiration from multi-view learning, we achieve information amalgamation by learning a common subspace. This involves projecting datasets from various spaces into the shared subspace $V$, thereby streamlining subsequent multi-label classification tasks. To foster greater consistency in the learned latent space representations $V_v$, we impose distribution-based constraints, a concept we delve into further in the subsequent discussion.

The feature alignment module stands as a pivotal component of our proposed NOMWM, particularly in scenarios involving datasets with substantial disparities in feature characteristics. Hence, we introduce the feature alignment module of NOMWM, grounded in a distribution-centric perspective. This viewpoint posits that different datasets inherently belong to distinct distributions, necessitating a holistic approach to fusion from a distributional standpoint.

In pursuit of aligning multiple datasets from a distributional perspective, we endeavor to minimize the separation between the centers of each dataset in the latent space. To this end, we leverage the center $\mu_v$ of each dataset $V_v$ to encapsulate its distribution, imposing constraints to bring these centers into closer proximity. This endeavor enhances the consistency of learned low-dimensional representations, thus facilitating subsequent classification tasks.

By leveraging the learning of a common subspace, we harness the diverse features present across samples to their fullest extent. This approach optimally utilizes the informative content of all samples, thereby enhancing the predictive capacity of the model.

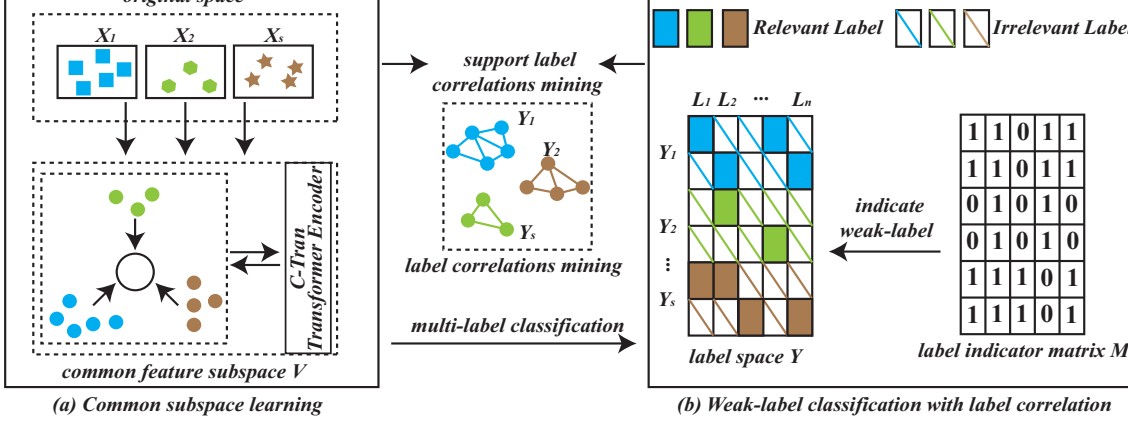

*(a) Common subspace learning*

*(b) Weak-label classification with label correlation*

**Figure 3: The specific process of our proposed NOMWM. It consists of the part of latent space representation learning subfigure(a) and the weak-label classification subfigure(b).**

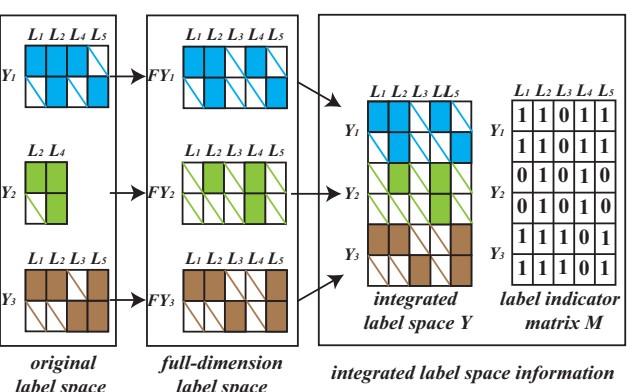

**Figure 4: The specific process of integrating all label sets. We expand these label space matrices $Y_v \in R^{m_v \times n_v}$ to full-dimension label matrix $FY_v \in R^{m_v \times n}$. Then the integrated label set $Y \in R^{m \times n}$ can be obtained by cascading the full-dimension label matrix $FY_v \in R^{m_v \times n}$, and the corresponding label indicator matrix $M$ can be obtained.**

Subsequently, we utilize the latent space representation as input to train the weak-label classification component. In summary, the loss incurred in latent space representation learning can be articulated as follows:

$$\Phi(X_v, V_v, U_v) = \sum_{v=1}^{s} \|X_v - V_v U_v\|_F^2 + \alpha \sum_{v=1}^{s} \sum_{k \neq v}^{s} \|\mu_v - \mu_k\|_2^2 \quad (2)$$

where $\mu_v$ is the center of data sets in latent space. $V_v$. $V_v \in R^{m_v \times d_c}$ and $U_v \in R^{d_c \times d_v}$ are the corresponding latent space representation and transform matrix. Besides, we have the parameter $\alpha$ to balance the losses of projection and alignment.

To mine the dependencies between features and labels for better common subspace, this study uses the Transformer encoder based on feature embedding and label embedding [10, 28]. In this study,

image feature embeddings are represented as $P$, each vector $p_i \in R_c^d$ of $P \in R^{d_c \times s}$ corresponds to the feature maps from each view of the image. The label embeddings, which are represented as $Q$, are denoted by $Q = \{q_1, q_2, \cdots, q_n\} \in R^{d_c \times n}$, with $n$ representing the number of label attributes and $d_c$ indicating the dimensionality of the word-embedding vector. We consider a set of embeddings $H = \{p_1, \ldots, p_s, q_1, \ldots, p_n\}$ as input for the Transformer encoder. With Transformers, the self-attention mechanism learns the importance, or weight, of embedding $h_j \in H$ with respect to $h_i \in H$. We input both feature embedding and label embedding information into the Transformer encoder to obtain the new representation.

$$\overline{H} = \text{softmax}\left(\frac{W^q H (W^k H)^T}{\sqrt{d}}\right) W^v H \quad (3)$$

$$h_i' = \text{ReLU}\left(\overline{h}_i W^r + b_1\right) W^o + b_2 \quad (4)$$

where $W^k$ is the key weight matrix, $W^q$ is the query weight matrix, $W^v$ is the value weight matrix, $W^r$ and $W^o$ are transformation matrices, and $b_1$ and $b_2$ are bias vectors. We denote the final output of the Transformer encoder as $H' = \{p_1', \ldots, p_s', q_1', \ldots, q_n'\}$. During the model learning process, $V_v$ in Eq.(2) will be obtained after transformer transformation, correspondingly, here the transformer network will be learned based on the loss and gradient passed back from the optimization process.

## 3.2 Weak-label classification with label correlation

In this section, we delve into our weak-label learning component, devised to address the challenge of weak-label instances stemming from diverse training datasets. We focus on two key aspects: the utilization of an indicator matrix to signify missing labels and the incorporation of label correlation mining inspired by [5] [18]. The central concept behind label correlation mining involves reconstructing the label space through instance and label similarity, thereby enhancing classification performance.

Real-world datasets within the 'same field' often exhibit varying label compositions. In other words, the label sets $Y_1, Y_2, \ldots, Y_s$ corresponding to these datasets may encompass different labels. Resolving the issue of inconsistent label sets constitutes a pivotal aspect of our multi-label sample augmentation strategy. To this end, we tackle this challenge in two steps. Firstly, we consolidate the label sets $Y_v \in \{0, 1\}^{m_v \times n_v}$ from different datasets to derive a unified label set $Y \in \{0, 1\}^{m \times n}$. Subsequently, we leverage this integrated label set $Y$ for weak-label classification.

Addressing the integration of all label sets, we adopt a straightforward yet effective approach. We identify all unique labels across the datasets and record their total count as $n$. Subsequently, we expand each label space matrix $Y_v \in \{0, 1\}^{m_v \times n_v}$ to a full-dimensional label matrix $FY_v \in \{0, 1\}^{m_v \times n}$. This involves filling the expanded label positions with 0, as illustrated by label $L_3$ in $FY_1$ (refer to Figure 4). By concatenating these full-dimensional label matrices $FY_v \in \{0, 1\}^{m_v \times n}$, we obtain the integrated label set $Y \in \{0, 1\}^{m \times n}$.

However, the integration of full-dimensional label sets $FY_v \in R^{m_v \times n}$ may lack precision across all datasets. Despite expanding the label space to encompass $n$ labels, not all corresponding datasets have been relabeled, resulting in potential omissions within the integrated label set $Y$. For instance, an instance from dataset $X_i$ may possess labels absent in its original label set $Y_i$ but present in another dataset $Y_j$. This integration method may inadvertently classify such labels as missing within the integrated set $Y$. To rectify this issue, we employ weak-label learning methodologies and label correlation mining in subsequent steps.

To address the impact of inaccuracies within the integrated label set $Y$, we propose leveraging weak-label learning alongside label correlations mining strategies. This approach encompasses two key facets. Firstly, we employ a missing indicator matrix $M$ to flag missing labels, enabling the reduction of their influence on the overall model training error. Secondly, we enhance the precision of weak-label learning by exploring label correlations. Drawing inspiration from [5], we incorporate a regularization term into our label correlation mining strategy, crafted with a view towards smoothness assumption. This regularization term encapsulates the smoothness of both instance and label similarities.

In addition to these strategies, we endeavor to capture the local geometric structures of the feature space by defining an undirected weighted graph for each dataset within $X_1, X_2, \ldots, X_s$. Herein, we denote the weight matrix (i.e., similarity matrix) for $X_v$ as $S_v = (s_{ij})m_v \times m_v$, where $s_{ij}$ signifies the similarity measure between samples $x_i$ and $x_j$. This similarity measure can be defined as:

$$s_{ij} = e^{-\frac{\|x_i - x_j\|^2}{\sigma^2}} \qquad (5)$$

where $\sigma^2$ is a parameter that can be adjusted.

In sum, the loss of weak-label classification with label correlation can be expressed as:

$$\Psi(V_v, W, Y, S_v, L) = \beta\|(VW - Y) \circ M\|_F^2$$
$$+ \gamma \sum_{v=1}^{s} \|V_v W - S_v V_v W L\|_F^2 \qquad (6)$$

In the above two subsections, we introduce *latent space representation learning* and *weak-label classification with label correlation*

respectively. According to the introduction of these two parts, the objective function Eq.(1) can now be detailed as follows:

$$\min_{W,V_v,U_v,L} \quad \sum_{v=1}^{s} \|X_v - V_v U_v\|_F^2 + \alpha \sum_{v=1}^{s} \sum_{k \neq v}^{s} \|\mu_v - \mu_k\|_2^2$$
$$+ \beta\|(VW - Y) \circ M\|_F^2 + \gamma \sum_{v=1}^{s} \|V_v W - S_v V_v W L\|_F^2 \qquad (7)$$

The proposed NOMWM consists of four different parts. The first part $\sum_{v=1}^{s} \|X_v - V_v U_v\|_F^2$ is the information fusion for different data sets, and the second part $\sum_{v=1}^{s} \sum_{k \neq v}^{s} \|\mu_v - \mu_k\|_2^2$ is to complete the feature alignment. These two parts correspond to Figure 3(a). The third part $\|(VW - Y) \circ M\|_F^2$ is to complete the weak label learning based on the label indication matrix $M$, and the last part $\sum_{v=1}^{s} \|V_v W - S_v V_v W L\|_F^2$ is the label correlation mining module. These two parts are illustrated in Figure 3(b). The specific optimization solution for the above model is presented as follows.

## 3.3 Optimization

For the model we built, we use the iterative minimization approach to optimize its loss function. When one of the variables is fixed, the original optimization problem will degenerate into a convex subproblem, which is convenient to solve. We adopt the MANOPT toolbox [1] to implement gradient descent with line search for the update of $W, V_v, U_v$ and $L$. The specific steps will be described as follows:

When $V_v, U_v$ and $L$ are fixed, the subproblem of Eq.(7) is convex to $W$, then the gradient corresponding to $W$ can be easily obtained and we use gradient descent to update it. Similarly, with $W, U_v$ and $L$ are fixed, the subproblem of Eq.(7) is convex to $V_v$, then the gradient corresponding to $V_v$ can be easily obtained and we use gradient descent to update it. With $W, V_v$ and $L$ are fixed, the subproblem of Eq.(7) is convex to $U_v$, then the gradient corresponding to $U_v$ can be easily obtained and we use gradient descent to update it.

When $V_v, U_v$ and $W$ are fixed, we have the following equation for $L$ by setting the derivative of Eq.(7) w.r.t $L$ to zero,

$$(\sum_{v=1}^{s} (S_v V_v W)'(S_v V_v W))L = \sum_{v=1}^{s} (S_v V_v W)' V_v W \qquad (8)$$

Then the closed-form solution for $L$ can be written as follows:

$$L = Z^+ \sum_{v=1}^{s} (S_v V_v W)' V_v W \qquad (9)$$

where $Z = \sum_{v=1}^{s} (S_v V_v W)'(S_v V_v W)$ and $Z^+$ indicates the pseudo inverse matrix of $Z$.

## 4 EXPERIMENTAL RESULT

In this section, we first introduce the data sets we choose and the reasons for choosing them; secondly, we introduce the relevant evaluation metrics; then we introduce the experimental settings. Finally, specific experiments and corresponding analysis are given.

## 4.1 Data sets

To validate the efficacy of our proposed Non-Overlapped Multi-View Weak-Label Learning Guided by Multiple Correlations, we

**Table 1: Data sets properties**

| Data sets | Domain | labels | Instances | Cardinality |
|-----------|--------|--------|-----------|-------------|
| Corel5k | image | 374 | 5000 | 3.5 |
| Mirflickr | image | 38 | 25000 | 4.7 |
| IAPRTC12 | image | 291 | 19627 | 5.7 |
| ESPGame | image | 268 | 20770 | 4.7 |
| MS-COCO | image | 80 | 122585 | 2.9 |

have chosen five benchmark datasets for experimentation, aiming to ascertain the effectiveness of our method. Table 1 provides an overview of the statistics pertaining to these datasets, namely Corel5k [6], IAPRTC12 [7], ESPGame [8], Mirflickr [9], and MS-COCO [12]. For feature extraction in our experiments, we employ various methodologies across these datasets, encompassing Colour [27], GIST [21], HOG [3], SIFT [19], LBP [20], and VGG [23]. The former five are traditional features, while the latter constitute deep features, all extensively utilized in the domain of image processing. The combination of these datasets and features serves to comprehensively evaluate the effectiveness of our proposed method across diverse scenarios, detailed further in the Experimental Settings section.

It is essential to underscore that our proposed method primarily addresses scenarios characterized by a relative insufficiency of training samples in multi-label classification. NOMWM leverages sample augmentation to tackle the challenge of integrating information from different datasets and handling missing labels in multi-label learning. However, due to the absence of established benchmark datasets directly applicable to validating our method, we resort to integrating and repurposing existing benchmark datasets to emulate varied datasets within the 'same field', as elaborated in the Experimental Settings section. Notably, the datasets utilized in our experiments predominantly revolve around images, owing to the ease of extracting diverse features for conducting simulation experiments. Nonetheless, it's crucial to highlight that the NOMWM method we propose is applicable across various domains beyond images, extending to fields like music, text, and beyond.

In the multi-label learning problem, since each sample may have multiple category labels at the same time, the single-label evaluation metrics which are commonly used in traditional supervised learning, such as accuracy, precision, and recall, cannot be directly used for the performance evaluation of the multi-label learning system. Therefore, researchers have successively proposed a series of multi-label evaluation metrics. Here we consider three evaluation metrics, *i.e.*, Macro-F1 and Micro-F1, which are widely used in multi-label learning to evaluate the prediction performance [30].

## 4.2 Experimental Settings

We propose a simulation experiment based on the division of single dataset to further validate our method. This approach leverages readily available data from four datasets (Corel5k, Mirflickr, IAPRTC12, and ESPGame), already partitioned into training and test sets. For each dataset, we further partition the training set into six sub-datasets and extract six different features: Colour, GIST,

HOG, SIFT, LBP, and VGG. These sub-datasets, each possessing distinct features, simulate six independent datasets within the 'same field' for our experiments.

Both sets of experiments encompass scenarios with full labels and weak labels. In the former, the focus is on multi-label training datasets with complete label sets, while in the latter, we randomly remove a portion of labels from the label matrix to simulate weak-label learning. Specifically, we randomly nullify 20% of label elements from each sample to generate weak-label datasets.

In our experiments, we compare our proposed approach with five state-of-the-art multi-label classification methods: Incomplete Multi-View Weak-Label Learning (iMVWL) [25], Hybrid Noise-Oriented Multi-label Learning (HNOML) [2], Low-Rank Multi-View Learning in Matrix Completion for Multi-Label Image Classification (lrMMC) [17], Expand globally, shrink locally: Discriminant Multi-Label Learning with Missing Labels (DM2L) [33], and Non-Aligned Incomplete Multi-view and Missing Multi-label Learning (NAIM$^3$L) [11].

In the experimental comparisons, iMVWL, lrMMC, and NAIM$^3$L focus on multi-view multi-label classification, with adaptations made to accommodate incomplete features by imputing average values. On the other hand, HNOML and DM2L are designed for multi-label classification with single-view data and incomplete label sets, where we incorporate the latent feature $V$ learned by our proposed method for experimentation. Parameter selection for these methods involves random holdout of one-fifth of training data for validation. We adopt predefined parameter settings from related works for iMVWL, HNOML, lrMMC, and DM2L, while selecting parameters $\alpha$, $\beta$, and $\gamma$ from the range $10^{-10}, 10^{-9}, \cdots, 10^{10}$ for our method. All experiments are conducted on a 64-bit Linux workstation with an Intel E5-2650 CPU and 256GB memory.

We evaluate NOMWM, NAIM$^3$L, iMVWL, HNOML, lrMMC, and DM2L on multi-label data. Tables 2 and 3 present the performance of multi-label classification under scenarios of weak and full labels across four datasets. These tables encompass three evaluation metrics: Macro-F1 and Micro-F1. The second column in each table denotes the features of the test datasets during prediction, reflecting the method's robustness across different feature sets. Our experimental results yield several noteworthy observations:

NOMWM consistently outperforms most state-of-the-art methods across the four datasets. For instance, under the full label scenario in Table 2 and 3, NOMWM achieves improvements of 2.15% (Micro-F1) and 1.55% (Macro-F1) on the IAPRTC12 dataset using the Gist feature. Under weak label conditions in Table 2 and 3, NOMWM enhances performance by 0.81% (Micro-F1) and 2.18% (Macro-F1) on the ESPGame dataset using the SIFT feature. These results validate the efficacy of our proposed method. Across Micro-F1 and Macro-F1 metrics, NOMWM consistently outperforms existing methods by 1%-3%, showcasing its overall superiority.

NOMWM is compared against NAIM$^3$L, iMVWL, HNOML, lrMMC, and DM2L. Among these methods, NAIM$^3$L, iMVWL, and lrMMC effectively handle multi-view datasets, while HNOML and DM2L utilize the latent feature $V$ derived from our method. Moreover, all four methods adeptly address weak-label scenarios. This comprehensive comparison underscores the superiority of our approach.

**Table 2: Quantitative results on validation sets on evaluation criteria Micro-F1, where the best ones are highlighted in bold.**

| Data sets | Feature | Results on validation sets with missing 20% labels | | | | | |
|---|---|---|---|---|---|---|---|
| | | NOMWM | NAIM$^3$L | iMVWL | HNOML | lrMMC | DM2L |
| Corel5k | Colour | 0.0320 | **0.0362** | 0.0273 | 0.0327 | 0.0285 | 0.0302 |
| | Gist | **0.0739** | 0.0715 | 0.0645 | 0.0546 | 0.0615 | 0.0518 |
| | HOG | **0.0779** | 0.0745 | 0.0717 | 0.0739 | 0.0694 | 0.0682 |
| | SIFT | **0.1095** | 0.1081 | 0.0686 | 0.0740 | 0.0856 | 0.0721 |
| | LBP | **0.0528** | 0.0472 | 0.0498 | 0.0332 | 0.0406 | 0.0525 |
| | VGG | **0.0958** | 0.0871 | 0.0768 | 0.0790 | 0.0620 | 0.0768 |
| IAPRTC12 | Colour | 0.0786 | **0.0947** | 0.0762 | 0.0830 | 0.0608 | 0.0712 |
| | Gist | **0.1370** | 0.1155 | 0.0808 | 0.0602 | 0.0732 | 0.0793 |
| | HOG | **0.1013** | 0.0923 | 0.0664 | 0.0786 | 0.0980 | 0.0836 |
| | SIFT | **0.1185** | 0.1053 | 0.0732 | 0.0840 | 0.0855 | 0.0722 |
| | LBP | 0.0801 | 0.0716 | 0.0883 | **0.0935** | 0.0752 | 0.0837 |
| | VGG | **0.2281** | 0.1936 | 0.1485 | 0.1656 | 0.1254 | 0.1389 |
| ESPGame | Colour | 0.0377 | **0.0516** | 0.0575 | 0.0513 | 0.0620 | 0.0621 |
| | Gist | **0.1039** | 0.0958 | 0.0840 | 0.0784 | 0.0549 | 0.0681 |
| | HOG | 0.0907 | **0.0964** | 0.0862 | 0.0727 | 0.0680 | 0.0773 |
| | SIFT | 0.0883 | 0.0811 | **0.0891** | 0.0735 | 0.0849 | 0.0739 |
| | LBP | **0.1180** | 0.1003 | 0.0714 | 0.0732 | 0.0711 | 0.0602 |
| | VGG | **0.1846** | 0.1748 | 0.1553 | 0.1658 | 0.1526 | 0.1464 |
| Mirflickr | Colour | **0.4273** | 0.4054 | 0.3682 | 0.3898 | 0.3127 | 0.3136 |
| | Gist | 0.3807 | **0.3992** | 0.3150 | 0.3043 | 0.3570 | 0.3290 |
| | HOG | **0.4658** | 0.4221 | 0.3054 | 0.3531 | 0.3828 | 0.3867 |
| | SIFT | **0.4387** | 0.4138 | 0.3648 | 0.3416 | 0.3342 | 0.3028 |
| | LBP | 0.4155 | 0.4032 | **0.4261** | 0.3497 | 0.3116 | 0.3195 |
| | VGG | **0.6084** | 0.5851 | 0.5760 | 0.5600 | 0.5698 | 0.5360 |
| Data sets | Feature | Results on validation sets with full labels | | | | | |
| | | NOMWM | NAIM$^3$L | iMVWL | HNOML | lrMMC | DM2L |
| Corel5k | Colour | **0.0547** | 0.0487 | 0.0404 | 0.0446 | 0.0414 | 0.0410 |
| | Gist | **0.0725** | 0.0702 | 0.0587 | 0.0601 | 0.0626 | 0.0596 |
| | HOG | **0.0938** | 0.0832 | 0.0740 | 0.0842 | 0.0739 | 0.0686 |
| | SIFT | 0.0855 | **0.0993** | 0.0622 | 0.0795 | 0.0739 | 0.0873 |
| | LBP | **0.0797** | 0.0756 | 0.0565 | 0.0519 | 0.0540 | 0.0683 |
| | VGG | **0.0913** | 0.0909 | 0.0810 | 0.0885 | 0.0639 | 0.0614 |
| IAPRTC12 | Colour | **0.0943** | 0.0851 | 0.0803 | 0.0654 | 0.0631 | 0.0754 |
| | Gist | **0.1082** | 0.0917 | 0.0995 | 0.0853 | 0.0769 | 0.0737 |
| | HOG | **0.0961** | 0.0854 | 0.0782 | 0.0815 | 0.0847 | 0.0819 |
| | SIFT | **0.1277** | 0.1012 | 0.0862 | 0.0918 | 0.0760 | 0.0862 |
| | LBP | 0.0883 | 0.0834 | **0.0902** | 0.0811 | 0.0821 | 0.0854 |
| | VGG | **0.2296** | 0.2062 | 0.1706 | 0.1983 | 0.1529 | 0.1760 |
| ESPGame | Colour | **0.0841** | 0.0810 | 0.0683 | 0.0605 | 0.0693 | 0.0712 |
| | Gist | **0.1042** | 0.0949 | 0.0861 | 0.0744 | 0.0630 | 0.0738 |
| | HOG | 0.0766 | 0.0868 | 0.0803 | **0.0880** | 0.0564 | 0.0650 |
| | SIFT | **0.1012** | 0.0853 | 0.0714 | 0.0792 | 0.0902 | 0.0820 |
| | LBP | **0.0982** | 0.0906 | 0.0723 | 0.0828 | 0.0758 | 0.0705 |
| | VGG | 0.1453 | **0.1836** | 0.1793 | 0.1682 | 0.1384 | 0.1527 |
| Mirflickr | Colour | **0.4584** | 0.4273 | 0.3737 | 0.3905 | 0.3200 | 0.3536 |
| | Gist | **0.4273** | 0.4104 | 0.3682 | 0.3668 | 0.3591 | 0.3687 |
| | HOG | **0.4792** | 0.4512 | 0.3750 | 0.3794 | 0.3895 | 0.3747 |
| | SIFT | **0.4455** | 0.4167 | 0.3809 | 0.3610 | 0.3549 | 0.3600 |
| | LBP | **0.4162** | 0.4014 | 0.3812 | 0.3994 | 0.3321 | 0.3656 |
| | VGG | **0.6352** | 0.5991 | 0.5813 | 0.6054 | 0.5885 | 0.5531 |

Our proposed sample augmentation method significantly enhances classification accuracy. Across most cases in the full label scenario, our method consistently outperforms others. By augmenting the model's adaptability to diverse features through data augmentation, NOMWM provides richer information for training the classification model. In weak label scenarios where missing labels disrupt the label set distribution, NOMWM's label indication matrix and label correlation mining module mitigate the impact, ensuring robust classification capabilities.

These observations collectively affirm the efficacy and versatility of our proposed NOMWM method in addressing multi-label learning challenges.

**Table 3: Quantitative results on validation sets on evaluation criteria Macro-F1, where the best ones are highlighted in bold.**

| Data sets | Feature | Results on validation sets with missing 20% labels | | | | | |
|---|---|---|---|---|---|---|---|
| | | NOMWM | NAIM$^3$L | iMVWL | HNOML | lrMMC | DM2L |
| Corel5k | Colour | **0.0430** | 0.0340 | 0.0334 | 0.0369 | 0.0312 | 0.0362 |
| | Gist | **0.0865** | 0.0746 | 0.0645 | 0.0643 | 0.0620 | 0.0749 |
| | HOG | **0.0998** | 0.0818 | 0.0896 | 0.0715 | 0.0830 | 0.0696 |
| | SIFT | **0.1145** | 0.1035 | 0.0733 | 0.0950 | 0.0798 | 0.0838 |
| | LBP | **0.0564** | 0.0415 | 0.0340 | 0.0416 | 0.0475 | 0.0513 |
| | VGG | **0.1047** | 0.0961 | 0.0738 | 0.0850 | 0.0790 | 0.0702 |
| IAPRTC12 | Colour | 0.0838 | **0.0853** | 0.0473 | 0.0749 | 0.0549 | 0.0656 |
| | Gist | **0.1130** | 0.0975 | 0.0602 | 0.0811 | 0.0712 | 0.0817 |
| | HOG | **0.1052** | 0.0951 | 0.0864 | 0.0882 | 0.0824 | 0.0716 |
| | SIFT | **0.1061** | 0.0946 | 0.0826 | 0.0816 | 0.0714 | 0.0834 |
| | LBP | 0.0846 | 0.0757 | 0.0570 | 0.0636 | 0.0816 | **0.0994** |
| | VGG | **0.2189** | 0.1736 | 0.1122 | 0.1560 | 0.1088 | 0.1203 |
| ESPGame | Colour | 0.0365 | **0.0549** | 0.0578 | 0.0499 | 0.0374 | 0.0514 |
| | Gist | **0.0948** | 0.0846 | 0.0720 | 0.0812 | 0.0792 | 0.0614 |
| | HOG | 0.1035 | 0.0907 | **0.1076** | 0.0693 | 0.0620 | 0.0744 |
| | SIFT | **0.0922** | 0.0860 | 0.0814 | 0.0870 | 0.0734 | 0.0733 |
| | LBP | **0.0994** | 0.0974 | 0.0544 | 0.0711 | 0.0717 | 0.0509 |
| | VGG | 0.1749 | 0.1669 | 0.1673 | **0.1752** | 0.1591 | 0.1406 |
| Mirflickr | Colour | **0.4090** | 0.3834 | 0.3974 | 0.3285 | 0.3215 | 0.3104 |
| | Gist | 0.3568 | 0.3665 | 0.3594 | 0.3277 | 0.3086 | **0.3756** |
| | HOG | **0.4512** | 0.4133 | 0.3087 | 0.3904 | 0.3921 | 0.3372 |
| | SIFT | **0.4185** | 0.3807 | 0.3240 | 0.3144 | 0.3877 | 0.3725 |
| | LBP | 0.3927 | 0.3885 | 0.3654 | 0.3681 | 0.3594 | **0.3988** |
| | VGG | **0.5872** | 0.5559 | 0.5060 | 0.5407 | 0.5174 | 0.5240 |
| Data sets | Feature | Results on validation sets with full labels | | | | | |
| | | NOMWM | NAIM$^3$L | iMVWL | HNOML | lrMMC | DM2L |
| Corel5k | Colour | **0.0614** | 0.0582 | 0.0464 | 0.0529 | 0.0435 | 0.0422 |
| | Gist | **0.0884** | 0.0731 | 0.0785 | 0.0680 | 0.0610 | 0.0617 |
| | HOG | **0.1037** | 0.0885 | 0.0516 | 0.0960 | 0.0716 | 0.0757 |
| | SIFT | 0.0764 | 0.0661 | 0.0762 | **0.1092** | 0.0890 | 0.0843 |
| | LBP | 0.0585 | 0.0612 | 0.0514 | **0.0623** | 0.0413 | 0.0445 |
| | VGG | **0.1092** | 0.0910 | 0.0759 | 0.0866 | 0.0871 | 0.0558 |
| IAPRTC12 | Colour | **0.1021** | 0.0885 | 0.0635 | 0.0840 | 0.0526 | 0.0708 |
| | Gist | **0.1082** | 0.0965 | 0.0893 | 0.0978 | 0.0894 | 0.0822 |
| | HOG | **0.1078** | 0.0825 | 0.0723 | 0.0902 | 0.0859 | 0.0759 |
| | SIFT | **0.1174** | 0.0956 | 0.0843 | 0.0859 | 0.0772 | 0.0809 |
| | LBP | **0.0886** | 0.0783 | 0.0635 | 0.0782 | 0.0668 | 0.0644 |
| | VGG | **0.2302** | 0.2027 | 0.1370 | 0.1725 | 0.1360 | 0.1647 |
| ESPGame | Colour | 0.0761 | **0.0826** | 0.0602 | 0.0656 | 0.0434 | 0.0506 |
| | Gist | **0.1049** | 0.0831 | 0.0766 | 0.0890 | 0.0767 | 0.0713 |
| | HOG | 0.0933 | 0.0867 | **0.1087** | 0.0751 | 0.0756 | 0.0650 |
| | SIFT | **0.1056** | 0.0839 | 0.0704 | 0.0532 | 0.0802 | 0.0980 |
| | LBP | **0.0981** | 0.0818 | 0.0598 | 0.0790 | 0.0798 | 0.0642 |
| | VGG | 0.1523 | 0.1409 | 0.1592 | **0.1858** | 0.1358 | 0.1503 |
| Mirflickr | Colour | **0.4608** | 0.4433 | 0.3741 | 0.4186 | 0.3647 | 0.3804 |
| | Gist | **0.4084** | 0.3824 | 0.3631 | 0.3540 | 0.3474 | 0.3256 |
| | HOG | **0.4671** | 0.4359 | 0.3825 | 0.3916 | 0.3856 | 0.3887 |
| | SIFT | 0.3981 | 0.3571 | 0.3737 | **0.4011** | 0.3961 | 0.3846 |
| | LBP | **0.4489** | 0.4154 | 0.3794 | 0.3696 | 0.3778 | 0.3408 |
| | VGG | **0.6194** | 0.5763 | 0.5218 | 0.5427 | 0.5440 | 0.5423 |

## 5 CONCLUSION

In this paper, we propose a novel multi-label learning framework, Non-Overlapped Multi-View Weak-Label Learning Guided by Multiple Correlations (NOMWM), to address the challenge of insufficient labeled training samples in multi-label learning scenarios. Our method effectively integrates datasets with diverse features within a unified field, facilitating joint learning for sample augmentation and enhancing model classification accuracy. Additionally, we introduce a weak-label module comprising the label missing indicator matrix and label correlation mining, which addresses incomplete label set matching between datasets, enhancing model practicality and robustness. Experimental results demonstrate that NOMWM outperforms state-of-the-art multi-label algorithms under both full-label and weak-label conditions.

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
