# OpenReview forum: "Non-Overlapped Multi-View Weak-Label Learning Guided by Multiple Correlations"
_acmmm.org/ACMMM/2024/Conference — MM2024 Poster_

### Official Review · Reviewer_EQ15 · 2024-05-22

**Rating:** 4
**Confidence:** 2

**Summary:**

This paper introduces a new method for multi-label classification, named Non-Overlapped Multi-View Weak-Label Learning (NOMWM), which effectively integrates datasets with diverse features guided by multiple correlations. This method uses sample augmentation within a shared domain to mitigate overfitting due to insufficient training samples, and incorporates a missing label indicator matrix and label correlation mining techniques to address the issue of incomplete labels. Experimental results demonstrate that this method outperforms existing state-of-the-art techniques on multiple benchmark datasets, highlighting its effectiveness and practical utility.

**Strengths:**

1．Innovativeness: The proposed Non-Overlapped Multi-View Weak-Label Learning (NOMWM) framework is an innovative approach to multi-label classification that opens new avenues for handling multi-view and weak-label issues by integrating features and labels from different datasets.
2．Methodological Strengths: The method enhances sample diversity and richness through the concept of a shared domain, effectively addressing incomplete labels while maintaining feature diversity. This approach significantly improves the model's prediction accuracy and robustness.

**Limitations:**

1．The article mentions that 𝑉𝑣 can be obtained through information propagation within a transformer, but in the optimization phase, 𝑉𝑣 is updated by fixing other variables. This juxtaposition of arguments can make the article's logic appear somewhat confused and obscure the intended role of the introduced transformer. Furthermore, the integration of deep networks into traditional methods seems to disrupt the model's coherence. I recommend that the author elaborately detail the motivation for introducing the transformer, as well as the specific problems it aims to solve.
2．Although the experiments are comprehensively conducted in the field of image processing, the application and effectiveness of the model in other domains, such as text or audio, have not been explored, leaving the generality of the method to be verified.
3．The paper is relatively weak in theoretical analysis, lacking an in-depth analysis of the model's convergence and generalization performance.
4．The objective function of the model is composed of four parts and involves three hyperparameters. Additionally, there are numerous variables that need to be updated. However, the paper does not analyze the complexity of the algorithm, nor does it compare the computation time with other methods. This raises the question of whether the complexity of the algorithm could become a limitation in practical applications. Furthermore, the paper does not provide a detailed analysis of the specific impact of the three hyperparameters on the performance, despite the relatively large number of hyperparameters involved.
5．The main idea of this paper is to integrate sample sets from different datasets within the same domain to construct a multi-label classifier, aiming to prevent overfitting. However, the issues encountered, such as alignment of different views, handling of missing labels, and characterization of label correlations, are all addressed using previously established loss functions. This gives the impression that the paper largely pieces together ideas from other researchers. Furthermore, based on experimental results, the proposed method does not significantly outperform traditional multi-label multi-view classification methods, with F1 scores on some datasets remaining particularly low.

**Suitability:**

2

---

### Official Review · Reviewer_nh9C · 2024-05-25

**Rating:** 3
**Confidence:** 3

**Summary:**

The paper presents a novel framework, Non-Overlapped Multi-View Weak-Label Learning Guided by Multiple Correlations (NOMWM), aimed at addressing the challenges of insufficient labeled samples and label inconsistencies across datasets in multi-label classification tasks. The approach integrates samples from various datasets within a 'same field' and employs techniques like a common subspace learning and weak-label learning to enhance classification accuracy. Extensive experiments demonstrate the superiority of NOMWM over existing methods

**Strengths:**

1. "Non-Overlapped Multi-View Weak-Label Learning" is a meaningful topic.

2. The authors give a lot of related work in this paper for helping readers understand the background.

**Limitations:**

1. Some citation error should be corrected, such as "AIMNet by Liu [14]" should be "AIMNet by Liu et al. [14]".

2. The notations and the problem definition are somewhat confusing, so it is suggested that the authors make specific explanations in this paper.

3. The evaluation metric contains only 'F1', as I know, there are many metrics for multi-label classification tasks.

4. Comparison methods are too outdated.

5. The construction of Section 4.2 is not clear.

**Suitability:**

3

---

### Official Review · Reviewer_96d4 · 2024-05-29

**Rating:** 4
**Confidence:** 3

**Summary:**

This work proposes a novel method for learning in non-overlapping mutli-view scenarios with weak labels. The method incorporates two components, (1) a representation learning technique that applies Transformer encoder to fuse information from multiple datasets into a common latent subspace and (2) weak-label learning technique using label-correlation to perform multi-label classification.

**Strengths:**

1. The objective is clearly explained.

2. The idea of developing an non-overlapping multi-view weak-label learning method is novel.

3. The paper writing is easy to follow and read.

**Limitations:**

However, the methodology is not clearly explained, and the experiments should be improved.

My comments are as follows:

1.	Regarding the motivation of this work, apart from preventing overfitting, is there any other reason (or necessity) for developing the proposed method?

2.	It is not clear to me how Transformer encoder is used to optimize the objective in equation (2). Please elaborate.

3.	In equation (7), the second loss term scales quadratically w.r.t. the number of views. How well do your method scale w.r.t. the number of views, and can this be shown in an experiment?

4.	What is the time complexity of the iterative optimization approach described in Section 3.3?

5.	In section 4, the claim that “NOMWM is applicable across various domains beyond images, extending to fields like music, text, and beyond” should be supported by experiments conducted on other data modalities.

6.	Can you elaborate on the potential real-world applications of NOMWM?

7.	There are typos in the current manuscript. E.g., “ocaen” in line 71. Please double-check.

8.	This is a suggestion. The results would be much more convincing if you can visualize the alignment of multiple datasets in the latent space. The current paper also lacks exhibits of qualitative results.

**Suitability:**

2

---

### Meta-Review · Area_Chair_ZAc5 · 2024-07-02

**Recommendation:** Accept (Poster)
**Confidence:** 4

**Metareview:**

This paper proposed a multi-label learning approach to tackle insufficient labeled samples and label inconsistencies across datasets. Features from different datasets are projected into a common space. Label correlation is employed to make multi-label classification predictions.

Pros:
- The paper presents a new approach to weak-label learning, leveraging non-overlapped multi-view data and guided by multiple correlations.

- The proposed method shows promise in improving learning performance with weak labels, supported by comprehensive experimental results.

Cons:

- Original submission only evaluated on image domain. After rebuttal, additional evaluations in text, image and audio modalities are evaluated to demonstrate the effectiveness.

- The introduced method has many hyper-parameters and may require substantial higher complexity which could limit practical applications. In the rebuttal, authors provided complexity analysis and hyper-parameter evaluations to address the concerns.

Given the effective rebuttal and the consensus among reviewers, I recommend acceptance with suggestions for minor revisions to enhance clarity and address practical concerns raised during the review process.